# Investigating the Antituberculosis Activity of Selected Commercial Essential Oils and Identification of Active Constituents Using a Biochemometrics Approach and In Silico Modeling

**DOI:** 10.3390/antibiotics11070948

**Published:** 2022-07-14

**Authors:** Katyna J. Boussamba-Digombou, Maxleene Sandasi, Guy P. Kamatou, Sandy van Vuuren, Rafal Sawicki, Zeynab Fakhar, Alvaro M. Viljoen

**Affiliations:** 1Department of Pharmaceutical Sciences, Faculty of Science, Tshwane University of Technology, Private Bag X680, Pretoria 0001, South Africa; kathysmile080@gmail.com (K.J.B.-D.); sandasim@tut.ac.za (M.S.); kamatougp@tut.ac.za (G.P.K.); zb.fakhar@gmail.com (Z.F.); 2SAMRC Herbal Drugs Research Unit, Department of Pharmaceutical Sciences, Faculty of Science, Tshwane University of Technology, Private Bag X680, Pretoria 0001, South Africa; 3Department of Pharmacy and Pharmacology, Faculty of Health Sciences, University of the Witwatersrand, 7 York Road, Parktown 2193, Johannesburg 2193, South Africa; sandy.vanvuuren@wits.ac.za; 4Department of Biochemistry and Biotechnology, Faculty of Pharmacy, Medical University of Lublin, Aleje Racławickie 1, 20-059 Lublin, Poland; rafal.sawicki@umlub.pl

**Keywords:** tuberculosis, essential oils, gas chromatography mass spectrometry, biochemometrics, minimum inhibitory concentration, antimycobacterial, molecular docking

## Abstract

Tuberculosis (TB) is a disease caused by *Mycobacterium tuberculosis* which has become prevalent due to the emergence of resistant *M. tuberculosis* strains. The use of essential oils (EOs) as potential anti-infective agents to treat microbial infections, including TB, offers promise due to their long historical use and low adverse effects. The current study aimed to investigate the in vitro anti-TB activity of 85 commercial EOs, and identify compounds responsible for the activity, using a biochemometrics approach. A microdilution assay was used to determine the antimycobacterial activity of the EOs towards some non-pathogenic *Mycobacterium* strains. In parallel, an Alamar blue assay was used to investigate antimycobacterial activity towards the pathogenic *M. tuberculosis* strain. Chemical profiling of the EOs was performed using gas chromatography-mass spectrometry (GC-MS) analysis. Biochemometrics filtered out putative biomarkers using orthogonal projections to latent structures discriminant analysis (OPLS-DA). In silico modeling was performed to identify potential therapeutic targets of the active biomarkers. Broad-spectrum antimycobacterial activity was observed for *Cinnamomum zeylanicum* (bark) (MICs = 1.00, 0.50, 0.25 and 0.008 mg/mL) and *Levisticum officinale* (MICs = 0.50, 0.5, 0.5 and 0.004 mg/mL) towards *M. smegmatis*, *M. fortuitum*, *M. gordonae* and *M. tuberculosis*, respectively. Biochemometrics predicted cinnamaldehyde, thymol and eugenol as putative biomarkers. Molecular docking demonstrated that cinnamaldehyde could serve as a scaffold for developing a novel class of antimicrobial compounds by targeting FtsZ and PknB from *M. tuberculosis*.

## 1. Introduction

Tuberculosis (TB) is a highly contagious respiratory disease caused by an aerobic rod-shaped bacterium, *Mycobacterium tuberculosis* [1]. *Mycobacterium tuberculosis* is transmitted through direct inhalation or ingestion of small aerosol droplets that contain the micro-organisms. The droplets are generated by an infected individual and transmitted primarily through sneezing and coughing. Symptoms of a TB infection include persistent cough, constant fatigue, loss of appetite and weight, fever, coughing up blood, night sweats, chest pain, shortness of breath and swollen lymph nodes [2]. Tuberculosis is regarded as a third-world disease and is prevalent in southeast Asia (44%), Africa (24%) and the western Pacific (18%), while there is lower incidence in the eastern Mediterranean (8%), USA (3%) and Europe (3%) [3]. According to the WHO, in 2018, approximately 10 million people suffered from TB (58% men, 32% women and 10% children) [4] and almost 87% of this population were from high–burden countries. First–line treatment in TB chemotherapy consists of isoniazid, ethambutol, pyrazinamide and rifampicin [5], while second–line treatment consists of aminoglycosides (e.g., streptomycin), polypeptides (e.g., capreomycin), fluoroquinolones (e.g., levofloxacin) and other, less–effective, drugs, such as thioamides (e.g., ethionamide) [6]. Despite the well-structured treatment programs and ready availability of anti-TB drugs, mortality rates remain high, ranking TB among the top 10 causes of death globally and the leading cause of death from a single infectious agent [3,7]. Poor treatment outcomes due to drug resistance have been identified as one of the major contributors to TB deaths.

Drug resistance has become a global challenge, particularly in the treatment of TB [7]. Over the years, multiple forms of resistance have emerged, namely multi–drug resistant (MDR-), extensively drug resistant (XDR-) and totally drug resistant (TDR-) TB. These resistant forms have required drastic changes in treatment strategy, leading to an increase in the number of drugs prescribed and longer treatment periods of up to two years. Tuberculosis patients experience serious side-effects with longer treatment periods and an increase in treatment-related costs, which leads to poor patient compliance. The WHO documented that MDR–TB represents approximately 3.3% of new TB cases and 20.5% of previously treated cases in 2015. Extensively drug–resistant tuberculosis (XDR–TB) cases have been reported in 105 countries and some cases of TDR–TB, which is resistant to first- and second-line anti–TB drugs [7]. The emergence of resistant *Mycobacterium* strains has prompted the search for new anti-TB drugs that are highly potent, with fewer side-effects and a low risk of engendering resistance. For this reason, natural products have attracted attention due to purported lower side-effect profiles and a long history of traditional use for the management of various diseases and infections. The use of essential oils (EOs) for their antimicrobial properties has the potential to treat drug-resistant TB.

Essential oils are groups of low molecular weight compounds extracted from the flowers, leaves, stalks, fruits and roots of aromatic plants, by steam or hydro-distillation [8]. Their use dates back to ancient civilization [9], with applications in pain management, wound care, respiratory tract complaints, aromatherapy and spiritual relaxation [10]. The use of EOs as antimicrobial agents may lead to a lower chance of developing microbial resistance, while their natural origin affords them a relatively high degree of safety and consumer confidence. Furthermore, their volatile nature makes them ideal for use through inhalation, which is the ideal delivery method for direct action to the lungs, which is the site of infection. Various EOs have been investigated against MDR-TB strains including *Rosmarinus officinalis*, *Origanum vulgare*, *Ocimum basilicum* and *Mentha piperita* [11]. Cold–pressed terpeneless (CPT) Valencia orange (*Citrus sinensis*) oil was demonstrated to be effective against a variety of *Mycobacterium* species, including *M. bovis*, *M. avium* complex and *M. tuberculosis* [12]. Essential oils of *Piper* species (*P. rivinoides*, *P. cernuum* and *P. diospyrifolium*) displayed moderate activity against *M. tuberculosis* H37Rv bacillus with MIC = 125 µg/mL [13]. Antimycobacterial activity of the EO of *Tetradenia riparia* and 6,7-dehydroroyleanone against *M. tuberculosis* H37Rv was reported, with MICs = 62.5 μg/mL and 31.2 μg/mL, respectively [14]. Several other studies have documented good antimycobacterial activities of various EOs; however, very few have gone further to identify the bioactive constituents.

The current study aimed to screen a range of commercially important essential oils for possible antimycobacterial activity. Furthermore, identification of potentially active antimycobacterial compounds in EOs was performed using biochemometrics and the activity of the compounds was validated. This biochemometric approach overcomes the challenges of active compound identification through conventional bio-assay guided fractionation, which is biased towards abundant and more dominant compounds. Biochemometrics can separate active and inactive compounds within complex chemical data matrices, including those occurring at very low levels, by employing statistical modeling tools to correlate secondary metabolite profiles with biological activity, subsequently revealing putative biomarkers [15]. The process is less time-consuming, unbiased and environmentally friendly and identifies multiple bioactive compounds from a single analysis. 

In the absence of available resources to perform in vitro enzymatic studies, in silico approaches, such as molecular docking, have emerged as important tools to identify biological targets with potential therapeutic activity. Docking analysis was considered for exploring potential targets to support our experimental observations. Our in silico study identified two targets from *M. tuberculosis*, namely FtsZ and PknB, for further lead compound-based essential oil development. The target FtsZ is essential for cell division in bacteria. Inhibition of the protein prevents proper formation of the divisome, which leads to filamentation and eventual cell death, making it an attractive target for antibiotic research [16,17,18,19,20,21]. Protein kinase B (PknB) from *M. tuberculosis*, with possible roles in several signaling pathways involved in cell division and metabolism, has an essential role in sustaining mycobacterial growth [22,23,24,25]. Recent findings showed that PknB is essential for both in vitro growth and survival of *M. tuberculosis* in the host [26,27,28,29,30]. 

## 2. Results

### 2.1. Antimycobacterial Activity of Essential Oils

The results obtained from the microdilution assay for the non-pathogenic strains revealed minimum inhibitory concentrations (MICs) ranging from 0.25 mg/mL to >8.00 mg/mL (Appendix A). The MIC values for the positive controls, rifampicin and ciprofloxacin, ranged from 0.004 µg/mL to 0.50 mg/mL, with rifampicin presenting the lowest concentration. *Mycobacterium fortuitum* was the most sensitive strain, with 42 EOs (48.2% of the test EOs) inhibiting growth at concentrations ≤ 1 mg/mL. Twenty-three EOs (27.1%) exhibited good activity towards *M. smegmatis*, while only 12 (14.1%) displayed good activity towards *M. gordonae*, making it the least susceptible of the non-pathogenic *Mycobacterium* strains tested in this assay. In total, eighteen EOs (21.2%) displayed antimycobacterial activity towards two strains, while three EOs (3.5%) were active towards all three *Mycobacterium* strains.

The results for *M. tuberculosis* showed that the strain had higher resistance towards the EOs. Among all the EOs tested, 79 (92.9%) yielded MICs greater than the maximum concentration tested (>256 µg/mL), indicating poor antimycobacterial activity towards this strain. Only three EOs (3.5%) displayed good activity; these included *Cinnamomum zeylanicum* (8.00 µg/mL), *C. cassia* (4.00 µg/mL) and *Levisticum officinale* (4.00 µg/mL). These three EOs also showed noteworthy activity in the microdilution assay towards the three non-pathogenic strains. Overall, the three EOs displaying broad-spectrum antimycobacterial activity towards all four pathogens tested were *L. officinale*, which was the most active EO towards all the pathogens, followed by *C. zeylanicum* and *C. cassia*.

### 2.2. Chemical Profiles of the Essential Oils

The major constituents of the 85 EOs were determined using GC-MS/FID and analyzed using a targeted approach. Monoterpenes were identified as major compounds in several EOs (Appendix A). Compounds identified in high concentrations included limonene, which was identified in 29 EOs, α-pinene in 20 EOs, 1,8-cineole in 19 EOs, γ-terpinene in 15 EOs, linalool and β-pinene in 11 EOs, and camphene and sabinene in 10 EOs. Limonene was identified as the major compound in all the *Citrus* spp., while α-pinene, camphene, limonene and β-pinene were identified in *Abies* spp. Principal component analysis was performed on the aligned GC-MS data to observe variation in the composition of the EOs. Figure 1 is a PCA scores plot where the EOs are distributed according to the degree of chemical variation and color-coded by species name. A high degree of variance in the EO profile was observed as broad scattering of the samples in the four quadrants. The model statistics showed that only 11.4% variation was modeled by the first 2PCs where PC1 = 6.3% (R2X = 0.063) and PC2 = 5.1% (R2X = 0.051). There were no strong outliers, as all the oils were within the model boundary (Hotelling’s T2 95%), except for *C. camphora* (CT linalool) (CNCL), a moderate outlier, which presented a somewhat distinct chemical profile.

### 2.3. Identification of Bioactive Compounds Using Biochemometrics

To correlate the antimycobacterial activity of the EOs to their chemical composition and subsequently determine the biomarkers responsible for the activity, four separate OPLS-DA models were constructed for *M. smegmatis*, *M. fortuitum*, *M. gordonae* and *M. tuberculosis*. A summary of the model statistics for the four OPLS–DA models is presented in Table 1. The cumulative chemical variation in the data (R^2^X_cum_), and the predictive ability (Q^2^_cum_) of the models, were less than 50% for three models, which may result in poor model performance; however, validation experiments for the single compounds were determined to confirm that the predictive ability of the models was good. 

The model for *M. smegmatis* was fitted with one predictive and two orthogonal components (A = 1 + 2) (Figure 2A). A total variance of 11.2% (R^2^X_cum_ = 0.11) in the X-matrix was used to compute the model and the total variation explained in the Y-matrix was 75.2% (R^2^Y_cum_ = 0.75). Clear separation of the active (green) and non-active (blue) EOs was observed along the predictive component (Pp1) with a modeled variation of 2.21% (Pp1 = 0.022). Identification of the biomarkers filtered from the active group was possible on the S-plot (modeled covariance (Pp1, *x*-axis) vs. modeled correlation (p(corr) [1], *y*-axis)), where the variables are indicated in green (Figure 2B). Each variable (a point on the S-plot) is an RTM pair extracted from the GC-MS data. Each statistically significant RTM pair that was highly correlated to active EOs (Class 1) could be observed in the upper right quadrant (green). Tentative identification of the RTM pairs was performed using the NIST^®^, Mass Finder^®^ and Flavor^®^ libraries. The biomarkers identified as common compounds occurring in the highly active EOs towards *M. smegmatis* were α-cedrene, thymol, geranyl acetate/butyrate, carvacrol, geraniol, benzyl acetate, α-calacorene, isophytol and 5-heptylcyclohexa-1,3-diene.

Figure 3A is the scores scatterplot for *M. fortuitum*, where separation of the active (green) from non-active (blue) EOs can be observed along the predictive component (Pp1), with 2.65% chemical variation related to separation of the two groups. The model fitted with one predictive and two orthogonal components (A = 1 + 2) was computed using 11.7% (R^2^X_cum_ = 0.11) of the total variance in the X-matrix and the total variation explained in the Y-matrix was 79.2% (R^2^Y_cum_ = 0.79). Tentative identification of the RTM pairs filtered in the S-plot (Figure 3B) revealed the biomarkers associated with the EOs active towards *M. fortuitum* to be linalool, geranial, geranyl butyrate, cinnamaldehyde, eugenol and thymol. Two compounds that were previously identified as biomarkers in Figure 3B, were also filtered out for *M. fortuitum*; these were identified as thymol and geranyl butyrate.

The OPLS-DA model constructed for *M. gordonae* was fitted with one predictive and one orthogonal component (A = 1 + 1) and computed using 7.8% (R^2^X_cum_ = 0.078) of the total variance in the X-matrix; the total variation explained in the Y-matrix was 58.5% (R^2^Y_cum_ = 0.585) (Figure 4A). The S-plot of the variables (Figure 4B) assisted the identification of biomarkers associated with good antimycobacterial activity as cinnamaldehyde, thymol, eugenol and terpinene-4-ol/β-caryophyllene. For a third time, thymol was identified as a bioactive compound in EOs active against *Mycobacterium* strains, while cinnamaldehyde and eugenol were identified for a second time.

Figure 5A shows a scores scatterplot for *M. tuberculosis*, where very few EOs classified as active (green) separate from the non-active EOs along the predictive component. A 3% chemical variation between these two classes of oils was attributed to the bioactivity of the EOs, according to the OPLS-DA model, which was fitted with one predictive and one orthogonal component (A = 1 + 1). The total variance in the X-data matrix used to compute the model was 7.99% (R^2^X_cum_ = 0.0799) and the total variation explained in the Y-matrix was 96.9% (R^2^Y_cum_ = 0.969). The S-plot-filtered RTM pairs (Figure 5B) led to the identification of four active biomarkers, namely, cinnamaldehyde, α-calacorene, 5-heptylcyclohexa-1,3-diene and cinnamyl acetate. Again, an overlap in the bioactive compounds was observed where cinnamaldehyde, α-calacorene and 5-heptylcyclohexa-1,3-diene were predicted to be active towards the *Mycobacterium* strains.

In total, 13 chemical compounds were predicted to have good antimycobacterial activity and these, in a mixture or individually, may be responsible for the observed antimycobacterial activity towards the four pathogens tested. Considering all the compounds predicted by the four models collectively, eugenol, thymol and cinnamaldehyde were frequently associated with active samples (oils). To validate the model predictions, a selection of the identified biomarkers (based on availability) were tested for their antimycobacterial activity. Good antimycobacterial activity of three compounds (cinnamaldehyde, thymol and eugenol) was confirmed, as demonstrated by low MIC values (<1.0 mg/mL) towards *M. fortuitum*, *M. gordonae* and *M. smegmatis* and equal to 256 µg/mL towards *M. tuberculosis*, except for cinnamaldehyde (8.00 µg/mL) (Table 2).

### 2.4. Molecular Docking Analysis

The superior level of antitubercular potential demonstrated by cinnamaldehyde in this study prompted us to perform molecular-docking studies to identify the potential targets for this scaffold and understand the key molecular mechanisms governing the inhibitory activity against *M. tuberculosis*. A comprehensive literature review, as well as a target prediction algorithm implemented on the CHEMBL database with 14,000 targets, resulted in two potential therapeutic targets, namely FtsZ and PknB proteins from *M. tuberculosis*.

#### 2.4.1. Cinnamaldehyde-FtsZ and Binding Modes Analysis

To explore the energetically favored binding modes of cinnamaldehyde and its corresponding degree of affinities upon complexation, docking analysis based on a reference model of three co-crystallized ligands accommodated in the active site and two allosteric sites of FtsZ was performed (Figure 6). Recently, an experimental study by Alnami et al. [26] revealed two novel allosteric pockets in FtsZ from *M. tuberculosis* that have not been identified before. Figure 6 and Figure 7 present 3D-structures of the experimentally identified active sites and two allosteric sites from crystallized FtsZ, in complex with GDP and 4-hydroxycumarin ligands, respectively [26].

The theoretical predictions from the molecular docking study were that cinnamaldehyde could successfully dock into the active site and the two known allosteric sites of *M. tuberculosis* FtsZ. The docked pose showed variation in the degree of affinity and the complexation was stabilized by formation of several bonded and non-bonded interactions, as shown in Figure 8 and Appendix A. The docking scores and the minimum energy for the formation of the complex between the ligand and the receptor (the glide energy) was observed to be negative which suggests these molecules could serve as a pertinent starting point for the rational design of drugs targeting *M. tuberculosis* FtsZ (see Table 3). The more negative value of the docking score and the glide energy signify good binding affinity of the ligand toward the target and vice versa. Table 3 presents the intermolecular interaction energy values (glide score, glide energy) from the docking calculation. The best docked conformation of cinnamaldehyde against the active site of FtsZ revealed that the molecule anchored at the same co-ordinates as observed for the native ligand with favorable interactions (see Figure 8). Its docking score was found to be −2.96 with an overall binding energy (glide energy) of −21.79 kcal/mol interacting with Thr-130 and Ala-183 residues from chain-A lining the active site through direct hydrogen bonding and π-alkyl interactions, respectively. Lys-55, Ala-39 and Leu-47 binding residues from chain-B were found to be engaged in the allosteric site of (414) via π-sigma and π-alkyl interactions with cinnamaldehyde imposing favorable binding affinities upon complexation. Similarly, Thr-306, Arg-304 and Ile 225, through hydrogen bonding formation and π-sigma and π-alkyl interactions, stabilized cinnamaldehyde in the second allosteric site (514). These types of hydrogen-bonding, and the π-sigma and π-alkyl interactions, served as an “anchor”, guiding the 3D orientation of cinnamaldehyde into the active site and two potential allosteric sites, facilitating ligand-binding upon complexation.

#### 2.4.2. Cinnamaldehyde-PknB Binding Mode Analysis

PknB from *M. tuberculosis* was identified among 14,000 available targets from the CHEMBL database using a target prediction algorithm implemented on its pipeline [27]. The generated potential targets using this pipeline are provided in Appendix A. The results present PknB as the only potential target for cinnamaldehyde with activity threshold: 6.5 [27]. The 3-D structure of mitoxantrone in the active site of PknB is provided in Figure 9 [22].

With a similar protocol as in Section 2.4.1, docking analysis, based on a reference model of co-crystallized mitoxantrone accommodated in the active site of PknB, was performed to obtain more insight into the binding mode of cinnamaldehyde upon PknB complexation. The results from the ensuing docking simulation revealed that cinnamaldehyde could fit snugly into the active site of PknB, occupying a position close to that of the native mitoxantrone ligand with a very similar topology of binding and varying magnitude of affinity. As depicted in Figure 10, cinnamaldehyde formed several network interactions with the active site of PknB through hydrogen bonding with Asp-156, π-sulfur interaction with Met-145 and three π-sulfur interactions with Leu-17, Ala-38, Met-155 in the binding site with a docking score of −5.57 and a glide energy of −20.94 kcal mol^−1^ (Table 4). Overall, it was evident from these docking simulations that cinnamaldehyde showed considerable affinity for both FtsZ and PknB targets from *M. tuberculosis*, which indicated that this scaffold represents a pertinent starting point for structure-based lead optimization.

## 3. Discussion

The chemical composition of an EO is directly related to its biological activity; therefore, there is a need for studies that consider this aspect when investigating the bioactivity of natural products. By correlating noteworthy antimycobacterial activity to EO profiles, the current study was able to identify active compounds in complex EO mixtures that can be explored further for development into antimycobacterial agents. The antimicrobial activity of some of these EOs and compounds have been reported previously. Studies of the bactericidal effect of low concentrations of *Eugenia caryophyllus* EO reported eugenol and eugenyl acetate as major constituents [28,29,30]. In the current study, eugenol was identified as the dominant compound in *E. caryophyllus* (Appendix A) which was active towards both *M. fortuitum* and *M. gordonae*. In a previous biochemometrics study, eugenol was also identified as a putative biomarker responsible for noteworthy antimicrobial activity towards a range of pathogens, namely, *Bacillus cereus*, *Staphylococcus aureus*, *Enterococcus faecalis*, *Escherichia coli*, *Pseudomonas aeruginosa*, *Candida albicans* and *Cryptococcus neoformans* [31]. A study investigating the antimicrobial activity of eugenol against carbapenem-resistant *Klebsiella pneumoniae* revealed eugenol’s ability to decrease microbial growth and intracellular ATP concentrations, and to induce changes in cell morphology and inhibit biofilm formation, at an MIC value of 200 μg/mL [32].

The EO of *Trachyspermum ammi*, with 42.3% thymol, was previously reported to demonstrate good antimicrobial and acaricidal activity in veterinary parasitology [33,34]. The current study identified thymol as a major compound in four EOs with good antimycobacterial activity, namely, *T. zygis* (Appendix A) and *T. ammi* (active towards *M. smegmatis* and *M. fortuitum*), *T. vulgaris* (CT thymol) (active towards *M. smegmatis* and *M. gordonae*) and *T. saturejoides* (active towards *M. gordonae*). Thymol was reported in many studies to have anti-oxidant, anti-inflammatory, local anesthetic, antinociceptive, cicatrizing, antiseptic, cardiovascular and antifungal properties [35,36,37,38,39,40,41,42,43,44,45]. In a study investigating the antimycobacterial properties of essential oil constituents, thymol was found to be the most active terpene, with MIC values of 0.78 and 2.02 μg/mL against strains of *M. tuberculosis* and *M. bovis,* respectively [46]. The antibacterial effect of thymol is suggested to be due to the disturbance of the lipid fraction of bacterial plasma membranes. This results in increased permeability of the membrane, inducing leakage of intracellular materials. In fungi, thymol causes enlargement of cell membranes, enabling passive diffusion of ions between the expanded phospholipids, thereby leading to disturbances in fungal biological functions [47]. 

The EO of *C. zeylanicum* bark displayed good antimycobacterial activity towards *M. smegmatis* and *M. tuberculosis*. One of the major constituents of cinnamon EO is cinnamaldehyde (Appendix A), which has been reported to be responsible for the strong antifungal and bactericidal activities of the EOs in previous studies. Eugenol, linalool, 1,8-cineole, neral and geranial were also reported to be responsible for the antibacterial activity of *Cinnamomum* EOs and to interact in a synergistic manner [48,49]. The trans-isomer of cinnamaldehyde with antimicrobial properties inhibits bacterial growth through various mechanisms of action [50], including alteration of the bacterial cell membrane [51], inhibition of ATPase [52], inhibition of cell division and separation [53,54], inhibition of membrane porins and more [55,56]. Other properties of cinnamaldehyde include anticancer/antitumor, cardiovascular, anti-inflammatory and as thermogenic agents [57].

Several experimental and modeling approaches have confirmed the inhibitory activity of cinnamaldehyde against FtsZ [58,59,60,61,62,63]. Li et al. [64] showed the in vitro antibacterial activity of cinnamaldehyde derivatives targeting FtsZ from different organisms, such as *Bacillus subtilis*, *E. coli*, *P. aeruginosa*, *S. aureus*, *Staphylococcus epidermidis* and *Streptococcus pyogenes*. Domadia et al. [54] demonstrated, through isothermal calorimetric titration and molecular docking, that FtsZ is one of the target proteins for cinnamaldehyde. Another study by Valero [27] showed the inhibitory effect of cinnamaldehyde on cell division via FtsZ, by significantly decreasing the exponential growth rate of *B. cereus* through interacting with the V208 and G295 binding site residues, corresponding to the V206 and G292 residues of FtsZ from *M. tuberculosis*. Experimental work undertaken by Kumar et al. [16] identified that zantrins inhibit FtsZ from both *M. tuberculosis* and *E. coli* with IC_50_ of 50–70 µM and 4–25 µM, respectively.

In an extensive in silico study, target identification prediction algorithms, implemented on the CHEMBL database with 14,000 available targets, were used to search for potential proteins for the cinnamaldehyde template. We were motivated to consider cinnamaldehyde as a template for FtsZ and PknB targets from *M. tuberculosis* for further study using molecular docking analysis. The case study in in silico molecular docking suggests FtsZ and PknB as potential targets from *M. tuberculosis* for cinnamaldehyde with considerable binding affinity energies. The predicted outcome presented here suggests a strong platform for the rational design of novel selective and potent *M. tuberculosis* inhibitors, based on a cinnamaldehyde scaffold.

## 4. Materials and Methods

### 4.1. Essential Oils, Chemical Reagents and Solutions

The EOs used were selected based on commercial significance, availability and chemical diversity. All 85 EOs were purchased from Pranarôm International (Belgium) and stored at 4 °C prior to analysis (Appendix A). The Middlebrook 7H9-S broth medium base used in the microdilution assay was purchased from Remel (USA), while the Middlebrook oleic albumin dextrose catalase (OADC), glycerol, the supplement albumin-dextrose-catalase (ADC), the *p*-iodonitrotetrazolium violet (INT) and Middlebrook 7H9 broth were purchased from Sigma-Aldrich (Germany). The Middlebrook 7H9-S broth medium base used in the microplate Alamar blue assay was purchased from Becton, Dickinson and Company (USA) and the Lowenstein–Jensen slopes were purchased from Thermo Fisher Scientific (USA). The microtiter plates used were purchased from Sigma-Aldrich (Germany). The positive control antibiotics, rifampicin, ciprofloxacin, ethambutol, streptomycin and isoniazid were supplied by Sigma-Aldrich, along with acetone, which was used as the negative control.

### 4.2. Mycobacterial Strains, Media and Culture Conditions

Three non-pathogenic laboratory reference strains of *Mycobacterium*, namely *M. smegmatis* ATCC 19420, *M. fortuitum* ATCC 6841 and *M. gordonae* ATCC 14470, were supplied by the Department of Pharmacy and Pharmacology, University of the Witwatersrand, South Africa. These strains belong to the Mycobacteriaceae family, known for a rapid growth rate with the ability to adapt to different environmental niches [65]. The strains were selected due to their unique cell-wall composition, which is remarkably thick and lipid-rich compared to other common pathogenic strains of *M. tuberculosis* [65]. The fourth, semi-pathogenic, *M. tuberculosis* H37Ra ATCC 25177 strain, was provided by the Department of Microbiology, Medical University of Lublin, Poland. The *M. smegmatis*, *M. fortuitum* and *M. gordonae* strains were maintained in 20% glycerol at −20 °C prior to use. *Mycobacterium smegmatis* and *M. fortuitum* were cultivated for approximately 3 to 4 days, while *M. gordonae* was cultivated for approximately 10 to 15 days at 37 °C. Following incubation, the bacterial cultures were standardized by dilution to 1:100 using Middlebrook broth with OADC growth supplement, to achieve approximately 1 × 10^6^ colony-forming units (CFU)/mL (0.50 McFarland turbidity standard) [66]. *Mycobacterium tuberculosis* H37Ra ATCC 25177 cultures were prepared on Lowenstein–Jensen slopes and incubated horizontally at 37 °C for two weeks. The bacterial suspension was then prepared by transferring the bacterial mass into 5 mL of Middlebrook 7H9-S broth media supplemented with 0.2% glycerol and 10% ADC. The suspension was vortexed for 2 min with 1 mm glass beads and incubated at room temperature for 1 hr to allow large clumps of microbial sedimentation. The supernatant (3 mL) was transferred and left to stand for another 30 min. This was then adjusted to the 0.5 McFarland turbidity standard with ADC supplemented Middlebrook 7H9 broth. The density of the bacterial suspension used was approximately 1 × 10^6^ CFU/mL [67].

### 4.3. Antimycobacterial Activity Determination

#### 4.3.1. The Microdilution Assay

The modified microdilution assay of Eloff [68] was used for MIC determination in a 96-well plate. Each EO was diluted with acetone to a 32 mg/mL stock. Acetone was included in the assay as a negative control to ensure it had no inhibitory activity towards the test mycobacterial strains; thus, the MIC values obtained would reflect the antimycobacterial activity of the EOs. The media was included as a growth control to demonstrate support for mycobacterial growth. Rifampicin and ciprofloxacin were used as positive controls in this assay. Initially, 100 µL of Middlebrook broth was added to all the microtiter plate wells. This was followed by adding 100 µL of each EO stock solution in duplicate in the first row of the plate. The growth control and acetone control were added in duplicate to the last four wells of the first row. The oils were then serially diluted. The control plates for each *Mycobacterium* strain were prepared in the same manner, using rifampicin (1.0 mg/mL) and ciprofloxacin (0.01 mg/mL). When all the test solutions and controls had been serially diluted across the microtiter plates, 100 µL of 0.5 McFarland standardized *Mycobacterium* cultures were added to the wells. The final EO concentrations in the wells of each column were 8, 4, 2, 1, 0.5, 0.25, 0.125 and 0.063 mg/mL [69]. The microtiter plates were sealed with sterile adhesive film to prevent EO loss due to their inherent volatility [70] and incubated at 37 °C for 4 days (*M. smegmatis* and *M. fortuitum*) and 10 days for *M. gordonae*. Following incubation, 40 µL of 0.4 mg/mL INT violet solution was added to each well and the plates were further incubated for 24 hrs, after which the MIC values were determined. All the wells with the purple–red coloration indicated mycobacterial growth, while the clear wells represented inhibition of mycobacterial growth. The lowest concentration of EO that prevented mycobacterial growth was taken to be the MIC value. All assays were undertaken at least in triplicate.

#### 4.3.2. The Alamar Blue Assay

Essential oil stock concentrations of 25.6 mg/mL were prepared by dissolving 5 µL of each EO in 190 µL of 2% dimethyl sulphoxide (DMSO). Each stock solution was diluted 1:50 using Middlebrook 7H9-S broth medium, resulting in a stock concentration of 512 µg/mL. An aliquot of 50 µL broth was added to each of the 96-wells in a microtiter plate and 50 µL DMSO was added to the control wells to give a final concentration of 1% DMSO for the negative control; 100 µL of stock EO was added to the test wells and finally 50 µL 0.50 McFarland standardized *M. tuberculosis* inoculum was added to each well. The final EO concentrations tested were 256, 128, 64, 32 and 16 µg/mL. A positive control plate was prepared in the same manner with the four antibiotics: ethambutol (128 µg/mL serially diluted to 0.25 µg/mL), streptomycin (64 µg/mL to 0.125 µg/mL), isoniazid (16 µg/mL to 0.03 µg/mL) and rifampicin (0.25 µg/mL to 0.0005 µg/mL). All the test and control plates were then sealed with sterile adhesive film and incubated at 37 °C for 7 days. Following incubation, 15 µL of Alamar blue was added to each well. The plates were resealed and further incubated for 48 hrs, before recording the MIC results. The change in color of the Alamar blue from blue to pink was assessed, and the fluorescence recorded on a spectrophotometer (570 nm/600 nm). The MIC value was confirmed by the lack of color change and defined as the lowest drug concentration that prevented blue to pink color change. Essential oils without color change beyond 16 µg/mL were repeated and the serial dilution was extended to 0.50 µg/mL [67]. All assays were undertaken at least in triplicate.

### 4.4. GC-MS Analysis of the Essential Oils

The 85 EOs were analyzed on an Agilent 6890 N GC system coupled directly to a 5973 MS (GC-MS/FID). Each EO was diluted with high grade hexane to a concentration of 20% *v*/*v*. A volume of 1 µL of each diluted EO was injected into the chamber and analyzed using a split ratio (200:1) with an autosampler at 24.79 psi and an inlet temperature of 250 °C. The GC system used was equipped with an HP-Innowax polyethylene glycol column of 60 m × 250 µm i.d. × 0.25 µm film thickness (Agilent Technologies, Hannover, Germany). The oven temperature was programmed as follows: 60 °C for the first 10 min, then rising to 220 °C at a rate of 4 °C/min and held for 10 min, and then rising to 240 °C at a rate of 1 °C/min. Helium was used as the carrier gas at a constant flow of 1.20 mL/min. Spectra were obtained on electron impact at 70 eV, scanning from *m*/*z* 35 to 550. The identification of the compounds was carried out using the NIST^®^, Mass Finder^®^ and Flavor^®^ databases, by comparing mass spectra, retention indices and authentic standards [71]. Quantification was performed using peak area normalization, where the peak areas obtained by FID for each compound was expressed as a percentage of the total of the peak areas of all the detected peaks.

### 4.5. Biochemometrics Analysis

To correlate the GC-MS profiles of the EOs to antimycobacterial activity, the two datasets were merged and chemometrics algorithms were employed to filter out biomarker molecules. Prior to the merger, chromatographic data pre-processing was performed in MarkerLynx^®^ v4.1 (Waters Corporation, Milford, MA, USA), which involved baseline correction, scaling, noise reduction and spectral alignment. The MS molecular fragments with the corresponding retention times were aligned across all the EOs. The aligned data of peak intensities were obtained in MS Excel^®^ and the MIC results were integrated with the GC-MS data. The EOs were assigned to classes based on the antimycobacterial activity, where class 1 EOs had MIC values ≤ 1.0 mg/mL (good activity) and class 2 EOs had MIC values > 1.0 mg/mL (poor activity) for the three non-pathogenic strains. The class allocation for the pathogenic *M. tuberculosis* was defined as MIC value ≤ 32 µg/mL (active EOs; Class 1) and MIC value > 32 µg/mL (non-active EOs; Class 2). The data was exported to SIMCA-P+ 14.0 (Umetrics, Sweden) for chemometric data analysis. Principal component analysis (PCA) was performed to provide an overview of the chemical variation in the dataset. Orthogonal projections to latent structures discriminant analysis (OPLS-DA) was applied to correlate the X-data matrix comprising of chromatographic data to the Y-variables created based on classification of antimycobacterial activity for each pathogen. Cross validation was applied to avoid statistically unreliable conclusions for group separation [72]. The models were constructed to separate chemical data related to antimycobacterial activity from other systematic variation. The results were visualized in a scores scatterplot where EOs with good activity were separated from those with poor activity. The S-plots were used to filter out putative biomarkers (EO compounds) responsible for the observed activity [31]. Retention time mass (RTM) pairs were filtered out in the S-plot and then compound identification was subsequently performed using NIST^®^, Mass Finder^®^ and Flavor^®^ libraries.

### 4.6. Antimycobacterial Validation Studies for Identified Active Compounds

The application of biochemometrics produced tentatively active compounds, predicted from the OPLS-DA model. In order to validate these findings, the active compounds were tested for antimycobacterial activity. The MIC value of each compound was determined using a microdilution assay towards *M. smegmatis*, *M. fortuitum* and *M. gordonae* and the Alamar blue assay towards *M. tuberculosis*.

### 4.7. Molecular Docking

The molecular docking study was performed using the standard protocol implemented in the grid-based ligand docking with energetics (GLIDE) module incorporated in the Schrodinger molecular modeling package to predict the binding modes of cinnamaldehyde in the active site of the FtsZ and PknB targets [73,74]. For this purpose, the X-ray crystal structures of the FtsZ complex (6Y1U.pdb) and the PknB complex (2FUM.pdb) were retrieved from the RCSB protein data bank (http://www.rcsb.org/pdb; accessed on 1 January 2000) and used as the primary model for the docking study [21,25]. The protein structure was refined for docking simulation using the protein preparation wizard incorporated in the GLIDE program [75]. This involved eliminating all crystallized water; as no water molecules were found to be conserved in the interaction with the protein, missing hydrogens/side chain atoms were added and the appropriate charge and protonation state was assigned to the protein structure corresponding to pH 7.0, considering the appropriate ionization states for the acidic, as well as basic, amino acid residues. The structure was then subjected to energy minimization using an OPLS-2005 force field with a root mean square deviation (RMSD) cut-off value of 0.30 Å to relieve steric clashes among the residues due to the addition of hydrogen atoms [76,77]. The three-dimensional structures of cinnamaldehyde were sketched with the build panel in Maestro and optimized using the ligprep module, which performs addition of hydrogens, adjusting realistic bond lengths and angles, correcting chiralities, ionization states, tautomers, stereo chemistries and ring conformations [78]. Partial charges were ascribed to the structures using the OPLS-2005 force-field. The resulting structures were then subjected to energy minimization until their average RMSD reached 0.001 Å. The active sites of the selected targets were defined for docking using the receptor grid generation panel which generates two cubical boxes having a common centroid to organize the calculations: a larger enclosing and a smaller binding box. The binding region was defined by a grid box with dimensions of 10 × 10 × 10 Å^3^ that was centered on the centroid of the native ligand in the crystal complex, which was sufficiently large to explore a larger region of the enzyme structure. Using this setup, automated docking was carried out to evaluate the binding affinities of the compounds within the macromolecule using the extra precision (XP) GLIDE scoring function to rank the docking poses and to measure their binding affinities. GLIDE searches for favorable interactions between the ligand and the active site of the enzyme using a filtering approach wherein each of the ligand poses pass through a series of hierarchical filters that evaluate the ligand’s interaction with the receptor. The docking poses of the ligands were visualized and analyzed using the Maestro’s Pose Viewer utility. The protocol adopted for docking simulation was validated by extracting the native ligand from the crystal structure and docking it into the active site of the selected targets using the above defined settings and monitoring its ability to reproduce the experimentally observed binding mode. The RMSD between the experimental conformation of the native ligand in the crystal structure and that obtained from its docking was found to be less than 1 Å, which confirmed the reliability of the docking procedure in reproducing the experimentally observed binding mode investigated here.

## 5. Conclusions

Using a biochemometrics approach, GC-MS and antimycobacterial activity data were successfully integrated and putative biomarkers responsible for the antimycobacterial activity were predicted by an OPLS-DA algorithm. The activity of three individual compounds, cinnamaldehyde, thymol and eugenol, was confirmed through further validation experiments. This study confirms the potential of the three compounds for further development into natural antimycobacterial agents, as previous studies have also documented the antimicrobial properties of these compounds. Furthermore, the study provides an overview of potential EOs that hold promise in inhibiting antimycobacterial growth. Molecular docking results obtained clearly indicate that cinnamaldehyde could be used as a platform for developing a new class of antimicrobial compounds to combat *M. tuberculosis* resistance by targeting FtsZ and PknB.

## Figures and Tables

**Figure 1 antibiotics-11-00948-f001:**
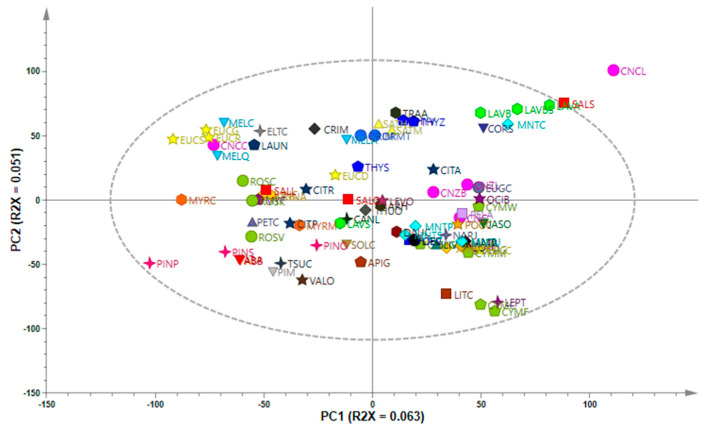
PCA scores scatterplot showing chemical variation in the essential oils based on GC-MS data.

**Figure 2 antibiotics-11-00948-f002:**
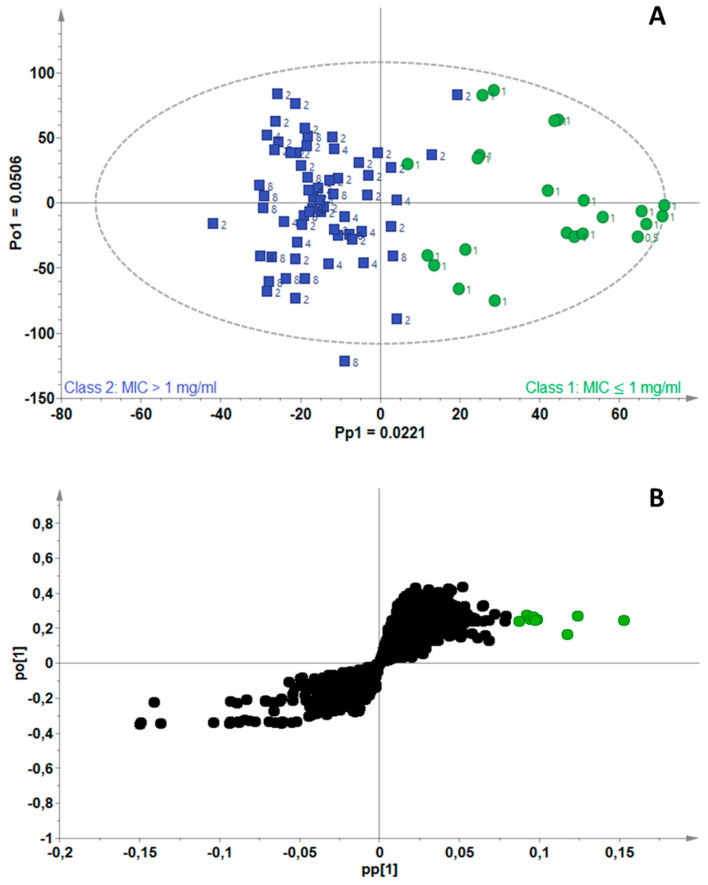
(**A**) OPLS-DA scores scatterplot for *Mycobacterium smegmatis*, showing separation of active (green) and non-active (blue) essential oils, (**B**) S-plot of the variables, indicating the putative biomarkers (green) associated with essential oils active against *M. smegmatis*.

**Figure 3 antibiotics-11-00948-f003:**
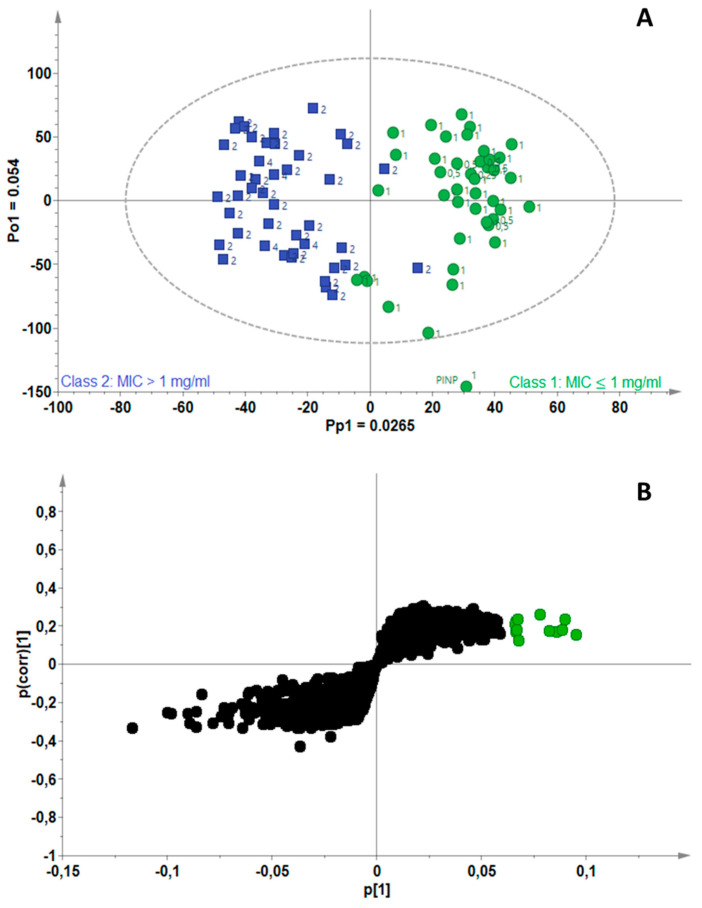
(**A**) OPLS-DA scores scatterplot for *Mycobacterium fortuitum,* showing separation of active (green) and non-active (blue) essential oils, (**B**) S-plot of the variables, indicating the putative biomarkers (green) associated with essential oils active against *M. fortuitum*.

**Figure 4 antibiotics-11-00948-f004:**
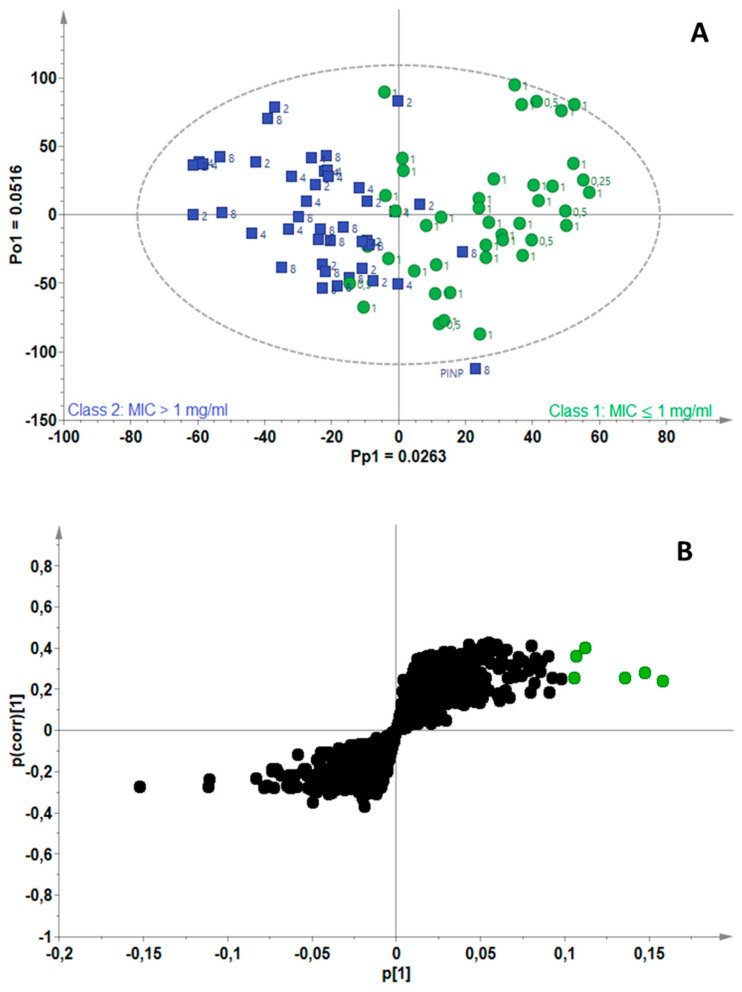
(**A**) OPLS-DA scores scatterplot for *Mycobacterium gordonae*, showing separation of active (green) and non-active (blue) essential oils, (**B**) S-plot of the variables, indicating the putative biomarkers (green) associated with essential oils active against *M. gordonae*.

**Figure 5 antibiotics-11-00948-f005:**
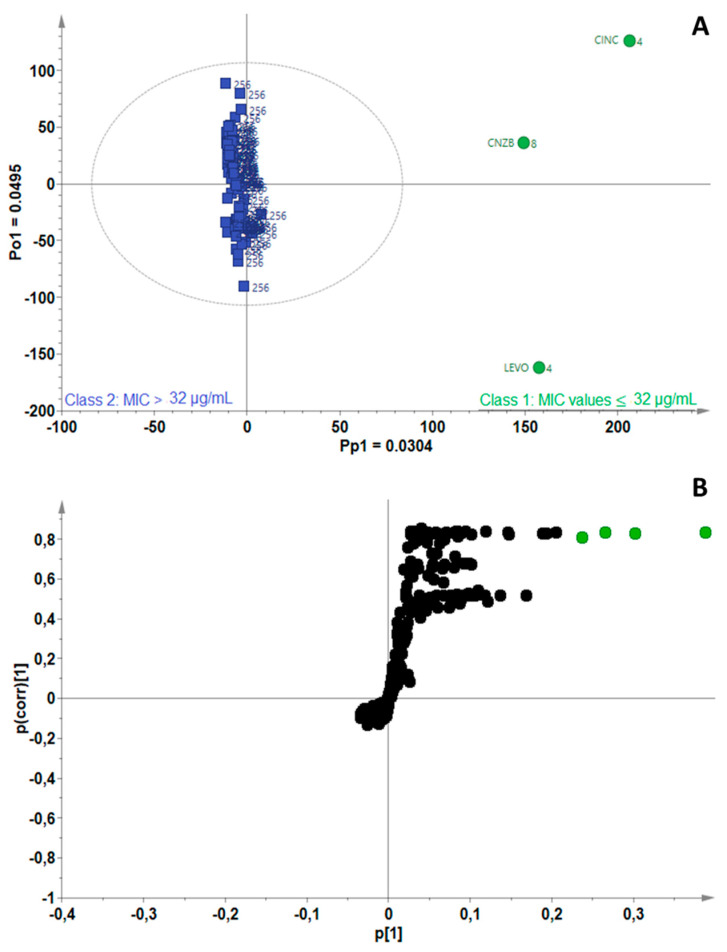
(**A**) OPLS-DA scores scatterplot for *Mycobacterium tuberculosis* showing separation of active (green) and non-active (blue) essential oils, (**B**) S-plot of the variables, indicating the putative biomarkers (green) associated with essential oils active against *M. tuberculosis*.

**Figure 6 antibiotics-11-00948-f006:**
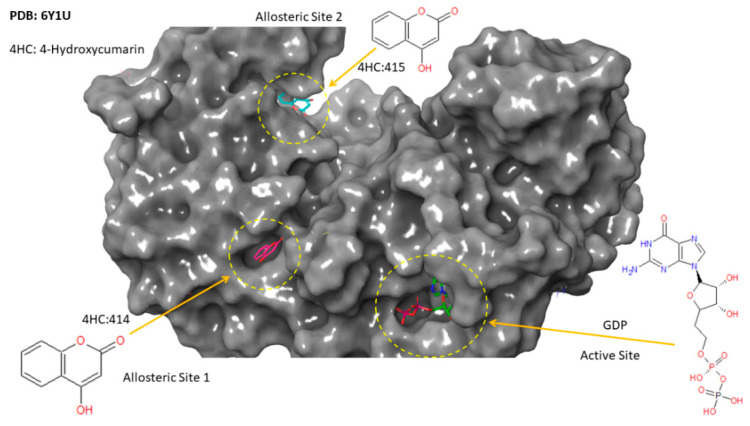
Surface view structure of FtsZ in complex with GDP nucleotide moiety in the active site and 4-hydroxycumarin ligand in two allosteric sites studied by Alnami et al. [26].

**Figure 7 antibiotics-11-00948-f007:**
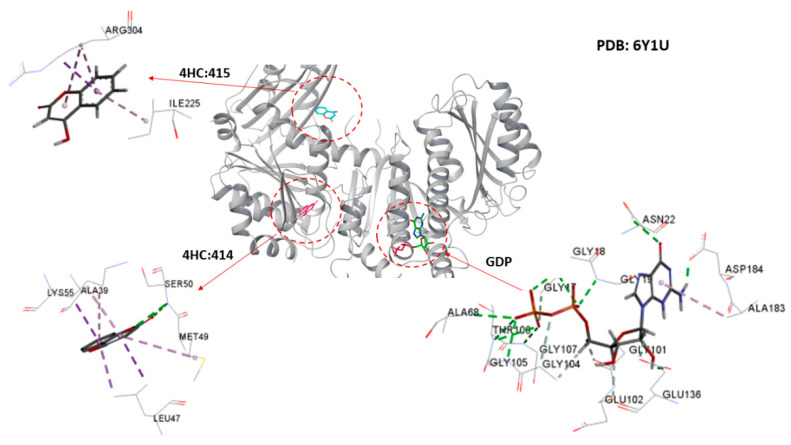
Binding site residues of FtsZ in complex with GDP nucleotide moiety and 4-hydroxycumarin ligand in the active site and two allosteric sites studied by Alnami et al. [26].

**Figure 8 antibiotics-11-00948-f008:**
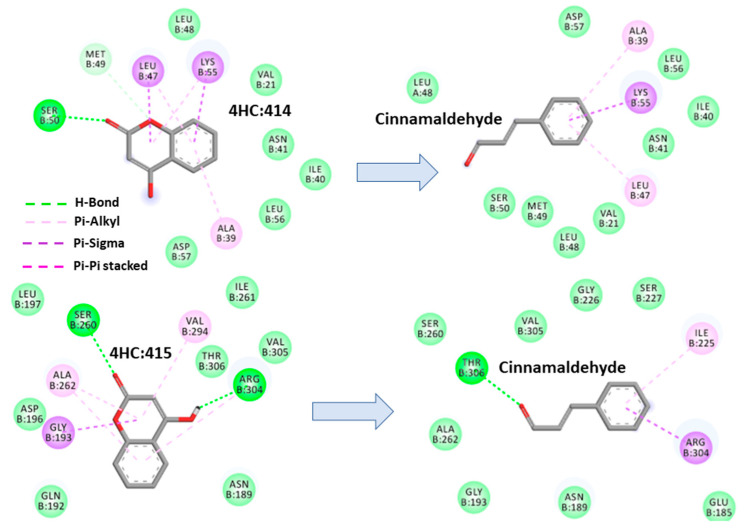
Binding modes for the best poses of the docked FtsZ complexes with 4-hydroxicumarin and cinnamaldehyde in the two allosteric sites.

**Figure 9 antibiotics-11-00948-f009:**
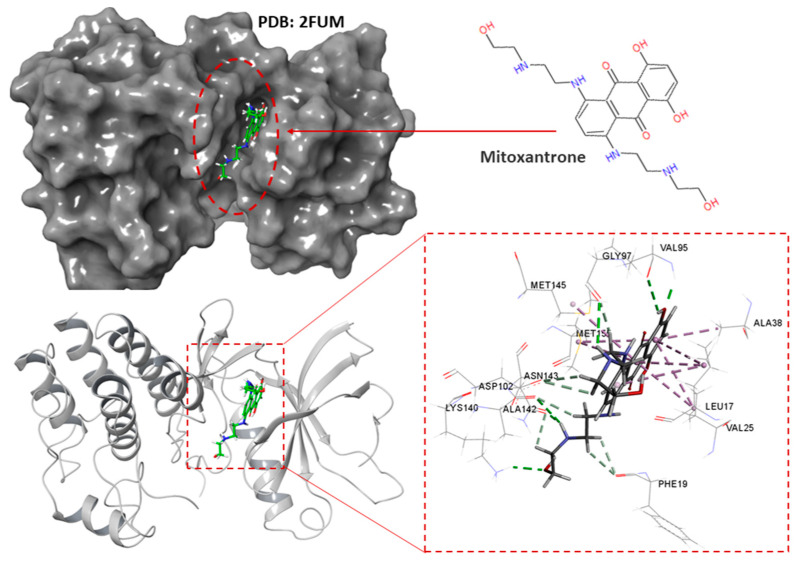
Surface view (**up**) and 3D structure view (**down**) of the co-crystalized mitoxantrone in the active site of PknB [22].

**Figure 10 antibiotics-11-00948-f010:**
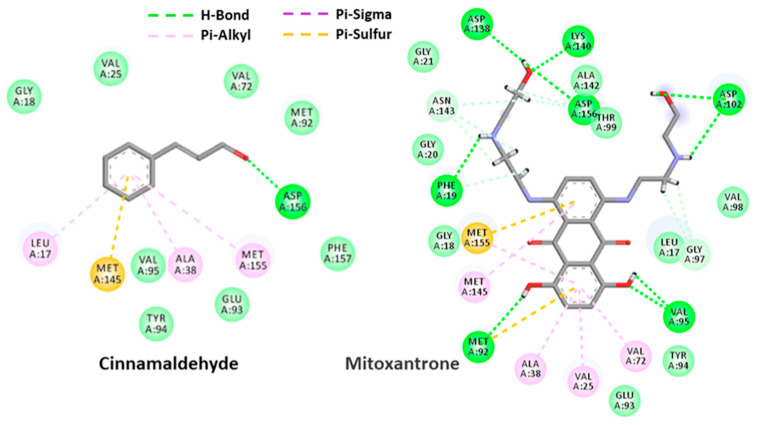
Surface view (**up**) and 3-D structure view (**down**) of co-crystalized mitoxantrone in the active site of PknB.

**Table 1 antibiotics-11-00948-t001:** OPLS–DA model statistics for the four *Mycobacterium* species.

*Mycobacterium* Strains	Number of Components (A)	R^2^X_cum_	R^2^Y_cum_	Q^2^_cum_
*M. fortuitum*	1 + 2	0.117	0.792	0.372
*M. gordonae*	1 + 1	0.078	0.585	−0.284
*M. smegmatis*	1 + 2	0.112	0.752	−0.115
*M. tuberculosis*	1 + 1	0.0799	0.969	0.577

**Table 2 antibiotics-11-00948-t002:** Mean MIC values (mg/mL) of predicted active compounds towards selected *Mycobacterium* strains.

Biomarker	*M. smegmatis* ATCC 19420	*M. fortuitum* ATCC 6841	*M. gordonae* ATCC 14470	*M. tuberculosis* H37Ra ATCC 25177 (µg/mL)
Cinnamaldehyde	0.250	0.130	0.030	8.00
Eugenol	1.000	1.000	0.250	256
Thymol	0.500	1.000	0.190	256
Ciprofloxacin (µg/mL)	0.310	0.160	0.310	-
Ethambutol (µg/mL)	-	-	-	2.00

**Table 3 antibiotics-11-00948-t003:** Summary of the binding affinities of co-crystalized GDP, 4-hydroxycumarin in two different allosteric sites and cinnamaldehyde in terms of docking scores and glide energies generated by GLIDE XP docking.

Entry	Compound	Glide Score. kcal mol^−1^	Glide Energy. kcal mol^−1^	Interacting Binding Site Residues
1	Co-crystal GDP	−9.04	−69.51	Asn-22, Thr-130, Arg-140, Asn-163, Phe-180
2	Co-crystal 4HC_414	−3.90	−21.37	Ala-39, Leu-47, Ser-50, Lys-55
3	4wCo-crystal 4HC_415	−4.21	−24.06	Gly-193, Ser-260, Ala-262, Val-294, Arg-304
4	Cinnamaldehyde active site	−2.96	−21.79	Ala-183, Thr-130
5	Cinnamaldehyde_4HC_414 site	−3.04	−21.88	Ala-39, Asn-41, Lys-55
6	Cinnamaldehyde_4HC_415 site	−3.16	−21.66	Ala-183, Thr-130

**Table 4 antibiotics-11-00948-t004:** Summary of the binding affinities of co-crystalized mitoxantrone and cinnamaldehyde molecules in terms of docking scores and glide energies generated by GLIDE XP docking.

Entry	Compound	Glide Score. kcal mol^−1^	Glide Energy. kcal mol^−1^	Interacting Binding Site Residues
1	Co-crystal mitoxantrone	−11.49	−64.21	Asn-22, Thr-130, Arg-140, Asn-163, Phe-180
2	Cinnamaldehyde	−5.57	−20.94	Ala-39, Leu-47, Ser-50, Lys-55

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
