# Peer review of "Investigating the Antituberculosis Activity of Selected Commercial Essential Oils and Identification of Active Constituents Using a Biochemometrics Approach and In Silico Modeling"

_antibiotics, 2022, doi:10.3390/antibiotics11070948_

Round 1
Reviewer 1 Report
In this manuscript, the authors are claimed to review the original article “Investigating the Antituberculosis Activity of Selected Commercial Essential Oils and Identification of Active Constituents Using a Biochemometrics Approach and in silico Modeling” with mentioned comments.
1. The MS have spelling errors such as line (4 & 27) which have two spelling of modelling/modeling.
2. Line no (271) corrects Figure no S1.
3. Line no (314) Figure 9 is not cited in the text.
4. Grammatical mistakes such as commas (before “and”), plural forms of words, and past tense must be strictly corrected in overall MS.
This MS requires minor grammatical correction. I recommend the submission of this MS with minor corrections in the article.
Author Response
Comment 1: The MS have spelling errors such as line (4 & 27) which have two spelling of modelling/modeling.
Response. The spelling for modelling is corrected throughout the manuscript.
Comment 2: Line no (271) corrects Figure no S1.
Response: Line no 271 is not related to Figure S1. Figure S1 has been mentioned in line 279, since this supplementary Figure is a part of Figure 8.
Comment 3. Line no (314) Figure 9 is not cited in the text
Response: Figure 9 is cited in line 310 before the image, we have highlighted the citation.
Comment 4: Grammatical mistakes such as commas (before “and”), plural forms of words, and past tense must be strictly corrected in overall MS.
Response: The manuscript was checked and all the commas that appeared before ‘and’ were removed. Furthermore, the manuscript was proof read for any further grammatical errors.
Comment 5: This MS requires minor grammatical correction.
Response: Thank you very much, the manuscript has been revised for grammar
Reviewer 2 Report
he paper by Boussamba-Digombou K. J. et al. reports antituberculosis activity of selected commercial essential oils and identification of active constituents using a biochemometrics approach and in silico modeling. The microdilution assay and the Alamar blue assay were used to determine the antimycobacterial activity of the EOs towards some non-pathogenic and the pathogenic M. tuberculosis strain, respectively. Chemical profiling of the EOs was performed using gas chromatography-mass spectrometry (GC-MS) analysis. Biochemometrics filtered-out putative biomarkers using orthogonal projections to latent structures-discriminant analysis (OPLS-DA). The potential therapeutic targets of the active compounds was predicted In silico modelling. Broad-spectrum antimycobacterial activity was observed for Cinnamomum zeylanicum (MICs = 1.00, 0.50, 0.25 mg/mL and 8.00 µg/mL) and Levisticum officinale (MICs = 0.50, 0.5, 0.5 mg/mL and 4 µg/mL) towards M. smegmatis, M. fortuitum, M. gordonae and M. tuberculosis, respectively. Molecular docking demonstrated that cinnamaldehyde could serve as a scaffold for developing a novel class of antimicrobial compounds by targeting FtsZ and PknB from M. tuberculosis. The paper is interesting and it may be published in antibiotics after addressing the following concerns.
1. The “biomarkers” in the abstract in line 32 should be changed to “hit-compounds”.
2. The units in abstract in line 29 and 30 should to be unified to same unit.
3. “M. fortuitum, M. gordonae and M. smegmatis and M. tuberculosis” in line 245 and 246 should write in italic form. And there are many similar mistakes in the manuscript, please carefully check.
4. The antimycobacterial unit of cinnamaldehyde in line 246 was different from it in Table 2, please carefully check.
5. Two references (J. Appl. Toxicol. 1984, 4, 283−292; Ann. Surg. 1894, 19, 102−111) about the important application of cinnamic acid as antitubercular agents should be added in the manuscript.
Author Response
Comment 1: The “biomarkers” in the abstract in line 32 should be changed to “hit-compounds”.
Response: The authors prefer to use the word ‘biomarkers’ as it is commonly used specifically when referring to biological activity related metabolomics studies for potentially active compounds. Please refer to evidence of such in the following DOI’s
doi:10.3390/molecules25163683
doi.org/10.1016/j.jep.2020.113681
doi. 10.1002/cbdv.201600218
doi.org/10.1016/j.chemolab.2013.11.004
Comment 2: The units in abstract in line 29 and 30 should to be unified to same unit.
Response: The units in the abstracts have been unified to mg/mL and the relevant conversions made:
8 µg/mL to 0.008 mg/mL
4 µg/ml to 0.004 mg/mL
Comment 3: M. fortuitum, M. gordonae, M. smegmatis and M. tuberculosis” in line 245 and 246 should write in italic form.
Response: Thank you. The species names have been corrected to italics
Comment 4: There are many similar mistakes in the manuscript, please carefully check.
Response: The manuscript was re-checked and the corrections made
Comment 5: The antimycobacterial unit of cinnamaldehyde in line 246 was different from it in Table 2, please carefully check.
Response: The µg/mL unit was added to the column heading as follows M. tuberculosis
H37Ra ATCC 25177 (µg/mL) to indicate that the values for this pathogen are in µg/mL, which matches with the text.
Comment 6. Two references (J. Appl. Toxicol. 1984, 4, 283−292; Ann. Surg. 1894, 19, 102−111) about the important application of cinnamic acid as antitubercular agents should be added in the manuscript.
Response: The manuscript currently carries 79 citations, which is a substantial number of references for a research article. There is also very little information in the manuscript relating to the work that we undertook.